# Updated values for Table 1 of fastest rubisco carboxylation rates in Davidi et al 2020

Yi-Chin Candace Tsai [ID][1], Zhijun Guo[1], Ron Milo [ID][2✉] & Oliver Mueller-Cajar [ID][1✉]

Comment on: D Davidi et al

In Davidi et al (2020), we performed a high-throughput spectrophotometric assay on over 100 rubiscos. The kinetic parameters of the seven fastest enzymes and two control enzymes were then characterized using radiometric 14-$CO_2$ fixation assays and active site quantification using the [14]C-labeled transition state analog carboxy-arabinitol 1,5-bisphosphate (CABP). Following the publication, we became aware that the spreadsheet used to compute the carboxylation velocities in the radiometric assay had not taken into account a twofold dilution of active sites in the carboxylation assay. Consequently, the carboxylation velocities measured using the radiometric assay and reported in Table 1 of Davidi et al were under-reported by a factor of two.

The correction of the reported values implies that the fastest enzyme from Gallionella sp. possesses a $k_{cat}$ value of ≈44 s$^{-1}$, which for the enzyme Rubisco appears even more extreme than the original values. To re-examine this extraordinary conclusion, we repeated the measurements and collected a new dataset, as shown in Fig. 1 (black symbols). This new dataset differed substantially from the published values after correcting for the factor 2 mistake (gray symbols in Fig. 1), prompting us to investigate the cause. Upon careful review of our laboratory records, we concluded that the CABP-binding assays used to quantify active sites in Davidi et al were inaccurate as described below. For most rubiscos, this inaccuracy manifested as an underestimation of active sites, and therefore overestimation of the carboxylation rate.

These first CABP-binding assays (referred to as Set 1) were performed in June/July 2019 on the same day as the carboxylation assays. Approximately 1 month later, prior to the original manuscript submission, we decided to ask whether the characterized enzymes were binding CABP stoichiometrically (Iniguez et al, 2021). We therefore collected a second set of CABP-binding assays (Set 2), and used this set to generate the published SDS-PAGE analysis and gel filtration chromatograms (Davidi et al, 2020, Appendix Figs. S4 and S5). As they were meant only for a stoichiometric analysis, no $CO_2$ fixation assays were performed in parallel to Set 2. Based on our new measurements, as we detail below, we now realize Set 2 was more representative.

If we assume that the quantifications achieved for Set 2 were accurate (as implied by Davidi et al, 2020, Appendix Figs. S4 and S5), we can multiply the $k_{cat}$ values of

Table 1. Comparison of kinetic data obtained in the current study with those reported in Davidi et al, 2020 after correcting for the missing factor of 2 and using CABP-binding assay Set 2.

| | Rate, $k_{cat}$ [s$^{-1}$] | | Affinity, $K_m$ ($CO_2$) [μM] | | Carboxylation efficiency, $k_{cat}/K_m$ [s$^{-1}$ mM$^{-1}$] | |
|---|---|---|---|---|---|---|
| | Davidi et al corrected | This study | Davidi et al corrected | This study | Davidi et al corrected | This study |
| Gallionella | 26.4 ± 1.1 | 27.5 ± 1.8 | 276 ± 25 | 253 ± 37 | 96 | 109 |
| Zetaproteobacterium | 27.3 ± 1.5 | 29.2 ± 1.1 | 261 ± 31 | 281 ± 23 | 105 | 104 |
| H. marinus | 24.0 ± 0.8 | 25.1 ± 0.6 | 162 ± 16 | 196 ± 11 | 148 | 128 |
| Sulfurivirga | 23.2 ± 0.6 | 21.4 ± 0.3 | 143 ± 10 | 161 ± 5 | 144 | 133 |
| Unknown organism (Actinosporangium) | 21.0 ± 0.9 | 19.7 ± 0.8 | 130 ± 17 | 166 ± 18 | 162 | 119 |
| Zetaproteobacteria_53_45 | 26.7 ± 1.6 | 26.9 ± 1.8 | 284 ± 35 | 300 ± 41 | 94 | 90 |
| A. ferrooxidans | 15.1 ± 0.7 | 17.1 ± 0.6 | 239 ± 27 | 282 ± 22 | 63 | 61 |
| R. rubrum | 11.0 ± 0.4 | 12.3 ± 0.5 | 109 ± 13 | 130 ± 16 | 101 | 95 |
| S. elongatus | 16.4 ± 0.7 | 14.2 ± 0.3 | 200 ± 20 | 156 ± 9 | 82 | 91 |

[1]School of Biological Sciences, Nanyang Technological University, 60 Nanyang Drive, 637551 Singapore, Singapore. [2]Department of Plant and Environmental Sciences, Weizmann Institute of Science, Rehovot, Israel. ✉E-mail: ron.milo@weizmann.ac.il; cajar@ntu.edu.sg
https://doi.org/10.1038/s44318-025-00419-y | Published online: 7 April 2025

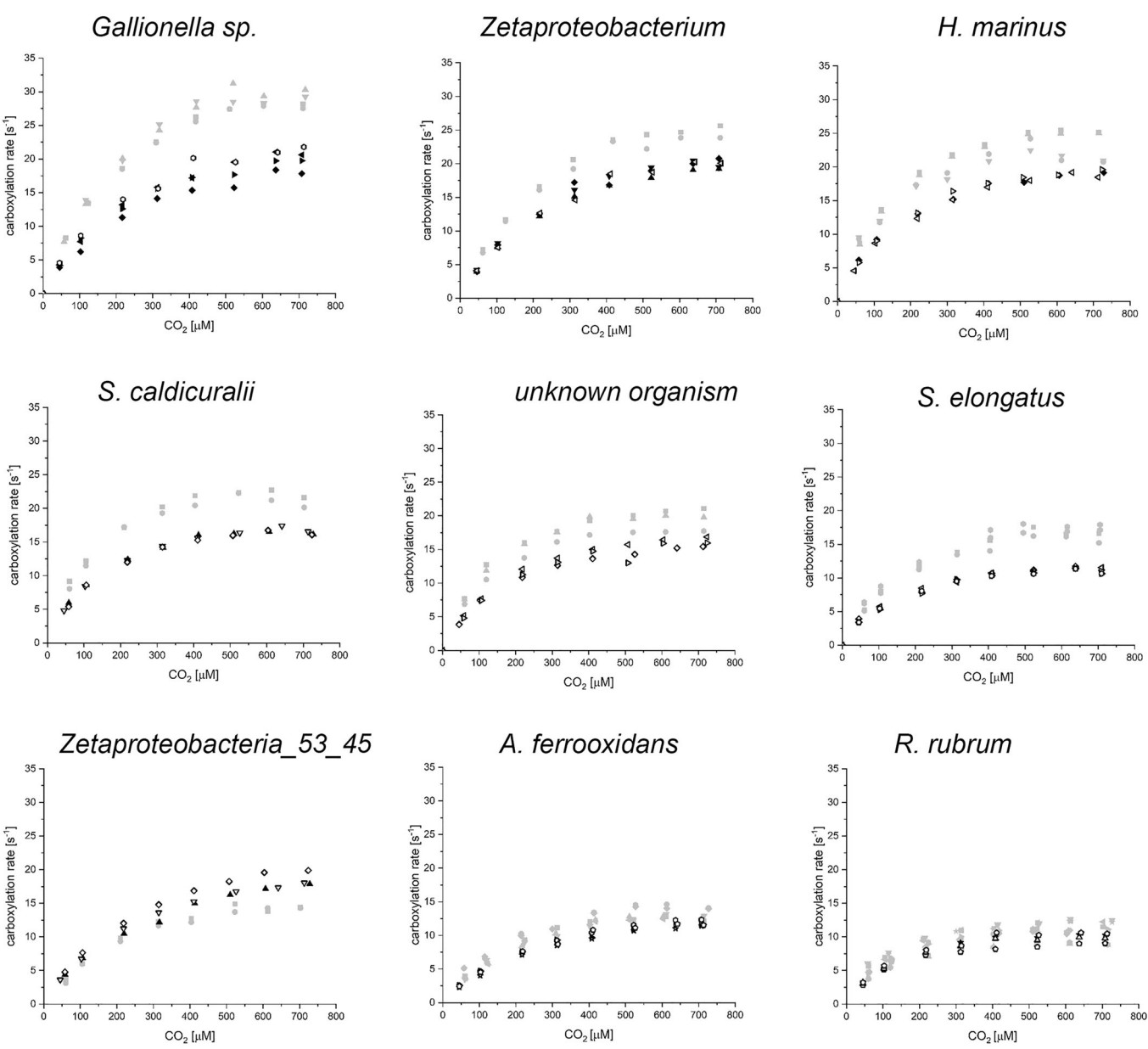

**Figure 1.  Comparison of the data points reported in Davidi et al (2020) after correcting for the factor two mistake (gray) and the newly collected dataset (black).**

Filled symbols were measured using protein purified in 2019, and open symbols were acquired using newly purified enzymes.

Davidi et al by the ratio of Rubisco stock concentrations determined in the two binding assays ([Set 1]/[Set 2]). This operation leads to turnover numbers that are within ≈15% of the newly generated dataset (see Table 1).

We also note that the original radiometric $k_{cat}$ values have a weak correlation with the spectrophotometric measurements used to rank the >100 Rubiscos in Davidi et al ($R^2 = 0.27$) (Fig. 2). Both the new dataset and the original dataset corrected for CABP binding using Set 2, now

show much higher correlation ($R^2 \approx 0.7$, Fig. 2).

Based on these arguments, we suggest to reject the Set 1 CABP-binding assays and correct the $k_{cat}$ values in Davidi et al's Table 1. The correction involves doubling the carboxylation rates and adjustment of active sites using the Set 2 CABP-binding data. The corrected values as well as the newly collected replicated dataset are shown in Table 1.

We are still uncertain why Set 1 resulted in undercounting of active sites for multiple

enzymes. We consider it likely that a technical factor that we are now unable to retrace in our records (or experimentally reproduce) was responsible.

As can be discerned from Fig. 2, the original active site quantification led to a strong relative overestimation of the $k_{cat}$ values for four metagenomic Rubiscos (*Gallionella*, *Zetaproteobacterium*, *H. marinus* and *S. caldicuralii*). Whereas *Gallionella* was previously pinpointed as the fastest Rubisco, now *Gallionella*, *Zetaproteobacteria* and *Zetaproteobacteria_53_45*

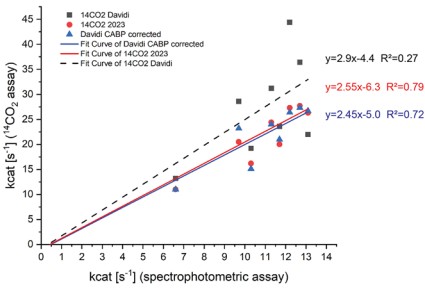

**Figure 2. Correlation between the *k_{cat}* values obtained using the coupled spectrophotometric Rubisco assay and the radiometric assay.**

Lower rates are expected for coupled spectrophotometric assays when compared to a direct assay, such as the radiometric discontinuous $^{14}CO_2$ fixation assay used here because of substrate limitation in the coupled reactions (Sharwood et al, 2016).

possess very similar $k_{cat}$ values between 26 and 29 s$^{-1}$. These velocities are two- to threefold faster than the *R. rubrum* $k_{cat}$ of 11–12 s$^{-1}$. We note the corrected *R. rubrum* $k_{cat}$ value is very similar to that reported in two earlier studies using an equivalent apparatus (Gomez-Fernandez et al, 2018; Mueller-Cajar et al, 2007). This value for *R. rubrum* $k_{cat}$ is higher than measurements in the literature that used approaches that are known to give slower rates as analyzed in (Iniguez et al, 2021).

In summary, our central finding, which reports the fastest Rubiscos to date, remains intact. We regret that a human error resulted in inaccurate carboxylation turnover rates being reported for the variants.

## Peer review information

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

## Acknowledgements

This work was funded by the Singapore Ministry of Education Academic Research Fund Tier 2 MOE-T2EP30120-0005 (to OM-C).

## Author contributions

**Yi-Chin Candace Tsai**: Conceptualization; Investigation; Methodology. **Zhijun Guo**: Investigation; Methodology. **Ron Milo**: Conceptualization; Data curation; Supervision; Investigation; Methodology; Writing—review and editing. **Oliver Mueller-Cajar**: Formal analysis; Supervision; Funding acquisition; Investigation; Methodology; Writing—original draft; Writing—review and editing.

## Disclosure and competing interests statement

The authors declare no competing interests.

