## [Peer Review File · The EMBO Journal]

Updated values for Table 1 of fastest rubisco carboxylation rates in Davidi et al. 2020

Yi-Chin Tsai, Zhijun Guo, Ron Milo, and Oliver Mueller-Cajar

Corresponding authors: Ron Milo (Ron.Milo@weizmann.ac.il) , Oliver Mueller-Cajar (cajar@ntu.edu.sg)

Review Timeline:

Submission Date:	7th Feb 25
Editorial Decision:	11th Feb 25
Revision Received:	6th Mar 25
Accepted:	7th Mar 25

Editor: Ieva Gailite

Transaction Report:

Dear Ron,

Thank you for submitting the updated dataset to the EMBO Journal. I have now received positive input from one of the original reviewers of Davidi et al. manuscript. Therefore, we will be happy to accept your article for publication as an author correspondence article after a few formatting points are addressed as listed below:

1. Please include author affiliations both in our online submission system and in the article text.
2. Please indicate the corresponding author and add their email in the manuscript text file.
3. All corresponding authors need an ORCID iD associated with their account. In order to link the ORCID iD to the account in our manuscript tracking system, the author in question has to do the following:
 - Click the 'Modify Profile' link at the bottom of your homepage in our system.
 - On the next page you will see a box halfway down the page titled ORCID*. Below this box is red text reading 'To Register/Link to ORCID, click here'. Please follow that link: you will be taken to ORCID where you can log in to your account (or create an account if you don't have one)
 - You will then be asked to authorise Wiley to access your ORCID information. Once you have approved the linking, you will be brought back to our manuscript system.Unfortunately, we cannot do this linking on the author's behalf for security reasons.
4. Please remove figures from the manuscript text file and upload as individual production quality figure files in the .eps, .tif, or .jpg format.
5. Please add a "Disclosure and competing interests statement" section (further info: <https://www.embopress.org/page/journal/14602075/authorguide#conflictsofinterest>).
6. Please add "Acknowledgments" section listing the funding sources. They should also be added and identical in our online system to ensure machine readability.
7. Please update references according to The EMBO Journal style - it should be alphabetically ordered; where there are more than 10 authors on a paper, the first 10 should be listed, followed by 'et al.' Please see further information here: <https://www.embopress.org/page/journal/14602075/authorguide#referencesformat>
8. Please move Figure legends after References.
9. Please move Table after Figure legends.

Please let me know if you have any questions regarding any of these points. You can use the link below to upload the revised files. I look forward to receiving the final version.

With best wishes,

Ieva

We realize that it is difficult to revise to a specific deadline. In the interest of protecting the conceptual advance provided by the work, we recommend a revision within 3 months (12th May 2025). Please discuss the revision progress ahead of this time with the editor if you require more time to complete the revisions.

Referee #1:

It is a credit to the authors that they have invested the time to properly troubleshoot perceived issues with their original kinetic data sets. The amount of work they have undertaken to do this is not trivial. The new values better accord with other 'high resolution' kinetic values in the literature (again, a testament of this team having the infrastructure and corporate knowledge to undertake such 'gold standard' assays for Rubisco). I therefore believe it critically important for these amended kinetics to be somehow made publicly available in association with the original publication.

The authors addressed the remaining formatting issues.

Dear Ron,

I am pleased to inform you that your manuscript has been accepted for publication in the EMBO Journal.

With best wishes,

Ieva
